# Could Long Non-Coding RNA MEG3 and PTENP1 Interact with miR-21 in the Pathogenesis of Non-Alcoholic Fatty Liver Disease?

**DOI:** 10.3390/biomedicines11020574

**Published:** 2023-02-15

**Authors:** Mustafa Genco Erdem, Ozge Unlu, Mehmet Demirci

**Affiliations:** 1Department of Internal Medicine, Faculty of Medicine, Beykent University, İstanbul 34398, Türkiye; 2Department of Medical Microbiology, Faculty of Medicine, Istanbul Atlas University, İstanbul 34403, Türkiye; 3Department of Medical Microbiology, Faculty of Medicine, Kirklareli University, Kırklareli 39100, Türkiye

**Keywords:** non-alcoholic fatty liver disease, lncRNA, miRNA, NAFLD

## Abstract

NAFLD is the most common cause of chronic liver disease worldwide. The miRNAs and lncRNAs are important endogenous ncRNAs families that can regulate molecular mechanisms. The aim of this study was to analyze the miRNA and lncRNA expression profiles in serum samples of NAFLD patients with different types of hepatosteatosis compared to healthy controls by the qPCR method. A total of180 NAFLD patients and 60 healthy controls were included. miRCURY LNA miRNA miRNome PCR human panel I + II kit and LncProfiler qPCR Array Kit were used to detect miRNA and lncRNA expression, respectively. DIANA miRPath and DIANA-lncBase web servers were used for interaction analysis. As a result, 75 miRNA and 24 lncRNA expression changes were determined. For miRNAs and lncRNAs, 30 and 5 were downregulated and 45 and 19 were upregulated, respectively. hsa-miR-21 was upregulated 2-fold whereas miR-197 was downregulated 0.25-fold. Among lncRNAs, NEAT1 was upregulated 2.9-fold while lncRNA MEG3 was downregulated 0.41-fold. A weak correlation was found between hsa-miR-122 and lncRNA MALAT1. As a conclusion, it is clear that lncRNA–miRNA interaction is involved in the molecular mechanisms of the emergence of NAFLD. The lncRNAs MEG3 and PTENP1 interacted with hsa-miR-21. It was thought that this interaction should be investigated as a biomarker for the development of NAFLD.

## 1. Introduction

Non-alcoholic fatty liver disease (NAFLD) is the most prevalent form of chronic liver disease globally, and the leading cause of chronic liver damage. It is a significant public health issue that is frequently overlooked [1,2]. NAFLD is a health condition in which imaging methods identify a fatty liver. Still, secondary factors such as substance abuse and alcoholism are not identified [3]. NAFLD is also connected with type 2 diabetes, metabolic syndrome, and obesity. Non-alcoholic steatohepatitis (NASH), a form of NAFLD, can also proceed to liver cirrhosis and hepatocellular cancer, resulting in severe complications that may necessitate liver transplantation [4]. While microRNAs (miRNAs) are known as short (20–24 nucleotides in length) endogenous non-coding RNAs (ncRNAs), long non-coding RNAs (lncRNAs) are defined as a family of endogenous ncRNAs with a length greater than 200 nt [5]. In recent years, breakthroughs in molecular methods have led to a vast increase in understanding regarding the impact of ncRNAs on human cells. It is known that miRNAs and lncRNAs are key RNA families that may influence biological processes, such as gene expression, but have little protein-coding activity. LncRNAs regulate gene expression during the transcriptional and post-transcriptional phases, whereas miRNAs target mRNA and regulate the gene expressions of agents that induce its degradation or inhibition [6]. Although it has been shown that the development of NAFLD is dependent on genetic, environmental, and metabolic variables, the pathogenesis process remains unclear. Additionally, how a simple fat deposition turns into a life-threatening, complex process is not yet understood. Although ncRNAs such as miRNA and lncRNA have been shown to have a role in the pathogenesis of NAFLD, it was predicted that the common interaction of these ncRNAs would help to explain the situation [7]. In the development of NAFLD, miRNAs are thought to be involved in metabolic and inflammatory processes and play an important role in the progression of the disease towards more severe stages. Furthermore, studies suggest that miRNAs influence the pathophysiology of NAFLD by directing the regulators that govern lipid metabolism, oxidative stress, and inflammation in the liver [8]. This study aimed to analyze the expression profile of 753 miRNAs and 90 lncRNAs in serum samples of NAFLD patients with different types of hepatosteatosis compared to healthy controls using the qPCR method, as well as the relationship between detected miRNA-LncRNAs and their association with regulatory pathways.

## 2. Materials and Methods

### 2.1. Study Design and NAFLD Diagnosis

In this observational study, 180 NAFLD patients and 60 healthy controls were included randomly. The collection and use of data and samples of NAFLD patients and healthy controls were approved by the ethics committee of the Istanbul MedicalPark Private Hospital (Approval no: 2021-1-7; Date: 26 April 2021). All participants provided their permission with full knowledge of the procedure. All subjects were diagnosed and graded for NAFLD utilizing abdominal ultrasonography exams based on the liver’s brightness and the presence of diffuse echogenicity in the liver parenchyma. Patients with NAFLD do not consume alcohol or specific medications that induce fatty liver. All subjects were negative for hepatitis conditions, including autoimmune hepatitis, HBV, HCV, and HDV. The NAFLD grade was recorded as none (0), mild (1), moderate (2), or severe (3) [9]. Inclusion criteria were participants who were aged >18 years, who used abdominal USG to evaluate hepatic steatosis grading (0 to 3), and who did not consume alcohol. Exclusion criteria were diagnosis with different hepatitis conditions, consumption of alcohol, positive results for HBsAg or Anti-HCV, presence of a cancer diagnosis, other chronic liver diseases or chronic infectious, a history of autoimmune disease or metabolic disease, and use of hepatotoxic medication.

### 2.2. Sample Collection

Blood samples were collected from every participant. Samples were immediately delivered to the laboratory in a chiller, followed by 10 min of centrifugation at 3000 rpm. In sterile microcentrifuge tubes, serum samples were separated and aliquoted. These aliquoted samples were kept at −80 degrees Celsius until RNA extraction.

### 2.3. miRNA Gene Expression Profilling Using qPCR Array

miRNeasy Serum/Plasma RNA isolation kit (Qiagen, Hilden, Germany) was used for miRNA isolation from the 200 µL of aliquoted serum samples. The DNase digestion protocol in the miRNeasy Serum/Plasma RNA isolation kit (Qiagen, Hilden, Germany) directive was applied during RNA isolations to avoid genomic DNA contamination. The miRCURY LNA RT Kit (Qiagen, Hilden, Germany) was used according to the manufacturer’s instructions to synthesize cDNA from 25 pg/L of extracted miRNAs. miRNA gene expression profiling was performed with the miR-CURY LNA miRNA miRNome PCR human panel I + II kit (YAHS-312Y) (Qiagen, Hilden, Germany). A pair of 384-well qPCR plates with predesigned PCR primer sets for specific miRNAs were included in the kit. The qPCR reaction was conducted using a LightCycler 480 (Roche Diagnostics, Mannheim, Germany) and a miRCURY LNA SYBR Green PCR Kit (Qiagen, Hilden, Germany) in accordance with the manufacturer’s instructions. qPCR conditions were 95 °C for 2 min for denaturation, followed by 40 cycles at 95 °C for 1 min and 56 °C for 1 min [10]. The SNORD38B, SNORD49A, and U6 genes were used as housekeeping genes. Livak and Schmittgen’s 2−CT (delta-delta Ct) method was utilized to detect specific miRNA gene expression levels through relative quantification. Relative ratio values were presented. A result of 0 to 1 was evaluated as downregulated, and >1 was evaluated as upregulated [11].

### 2.4. LncRNA Expression Profiling Using qPCR Array

RNA was extracted from 200 µL of aliquoted serum samples using the miRNeasy Serum/Plasma RNA isolation kit (Qiagen, Hilden, Germany). Then, 25 pg/µL of RNA was used for cDNA synthesis, which was performed using the RT2 First Strand Kit (Qiagen, Hilden, Germany) according to the manufacturer’s instructions. The LncProfiler qPCR Array Kit (System Biosciences (SBI), Palo Alto, CA, USA) was used to conduct LncRNA profiling analysis. The kit was provided with one 96-well ready-to-use qPCR plate containing predesigned PCR primer sets for specific lncRNAs. The qPCR reaction was performed on a LightCycler 480 (Roche Diagnostics, Mannheim, Germany) instrument using a LightCycler 480 SYBR Green Master Kit (Roche Diagnostics, Mannheim, Germany) according to the manufacturer’s instructions. The 18S rRNA, RNU43, GAPDH, LAMIN A/C, and U6 genes were used as housekeeping genes. Relative quantification analysis was used to detect specific lncRNA profiling and then analyzed by Livak and Schmittgen’s 2^−ΔΔCT^ (delta-delta Ct) method. Relative ratio values were presented. A result of 0 to 1 was evaluated as downregulated and >1 was evaluated as upregulated [12].

### 2.5. miRNA and Pathway Interaction Analysis by DIANA-miRPath Web Server

DIANA miRPath v3.0, a web server (https://dianalab.e-ce.uth.gr/html/mirpathv3/index.php?r=mirpath, accessed on 15 December 2022) was used for the analysis between the target miRNA and Kyoto Encyclopedia of Genes and Genomes (KEGG) pathways. This was an in silico miRNA target prediction algorithm [13].

LncRNA and miRNA connection analysis was conducted by the DIANA-lncBase web server.

DIANA-lncBase v3.0, a web server (https://diana.e-ce.uth.gr/lncbasev3/home, accessed on 15 December 2022) was used to conduct connection analysis between the miRNAs and lncRNAs.

### 2.6. Statistical Analysis

IBM SPSS version 20.0 software (SPSS Inc., Chicago, IL, USA) was used to perform statistical analysis. Data are presented as mean ± standard deviation. The Kruskal–Wallis test, Mann–Whitney U test and Spearman correlation analysis were used. *p* values below 0.05 (*p* < 0.05) were considered statistically significant. For Spearman’s analysis, which was used for correlation and values (r_s_), r_s_ < 0.25 was evaluated as not statistically correlated, r_s_ = 0.25–0.5 was evaluated as a weak correlation, r_s_ = 0.5–0.75 was evaluated as a moderate correlation, r_s_ = 0.76–0.85 was evaluated as a strong correlation, and r_s_ > 0.85 was evaluated as a very strong correlation [13].

## 3. Results

Table 1 provides demographic information for the 180 NAFLD patients and 60 healthy controls included in our study. The mean age of the NAFLD patients in our study was 39.09 ± 11.91 years, while the mean age of the healthy controls was 38.82 ± 8.5 years (Table 1).

NAFLD patients were graded according to ultrasound examinations, and their demographic data are presented in Table 2.

A total of 75 of 753 miRNA alterations in NAFLD patients were found to be significant compared to healthy controls (*p* < 0.05) by qPCR. Of these 75, 30 miRNAs were downregulated and 45 were upregulated compared to healthy controls. When lncRNA expression was examined, it was determined that 24 of 90 lncRNAs were altered significantly (*p* < 0.05); 19 of 24 lncRNAs were upregulated and 5 were downregulated (Figure 1).

Among the upregulated miRNAs, hsa-miR-21 showed the highest increase, at 2.3 ± 0.22 fold, followed by hsa-miR-222, hsa-miR-122, hsa-miR-10b, hsa-miR-221, hsa-miR-155, hsa-miR-33a, hsa-miR-223, hsa-miR-378a, and hsa-miR-193a, respectively. Among the downregulated miRNAs, hsa-miR-197 showed the greatest decrease, at 0.24 ± 0.07 fold, followed by hsa-miR-129, hsa-miR-99a, hsa-miR-422a, hsa-miR-27a, hsa-miR-139, hsa-miR-29a, hsa-miR-146b, hsa-miR-99b, and hsa-miR-296.

Among lncRNAs, only NEAT1 (family) (2.87 ± 0.23), MALAT1 (2.51 ± 0.31), MEG3 (family) (0.41 ± 0.12), and PTENP1 (0.48 ± 0.05) were detected in all samples. While NEAT1 (family) (2.87 ± 0.23) and MALAT1 (2.51 ± 0.31) were detected as upregulated, MEG3 (family) (0.41 ± 0.12) and PTENP1 (0.48 ± 0.05) were found to be downregulated. HOTAIR, EgoA, and Nespas, which were found to be upregulated in NAFLD patients, were found to be expressed in 114 (63.34%), 97 (53.88%), and 91 (50.56%) samples, respectively. MEG9, which was found to be downregulated in NAFLD patients, was found to be expressed in 93 (51.67%) samples. In healthy controls, HOTAIR, EgoA, Nespas, and MEG9 were detected in 34 (56.67%), 24 (40%), 28 (46.67%), and 38 (63.34%) samples, respectively.

According to in silico analysis conducted with the DIANA miRPath v3.0 online software, miRNAs that were found to be upregulated at the highest rates and downregulated at the lowest rates in our study had the greatest statistical impact on the “Fatty acid biosynthesis” pathway. Following this analysis, it was discovered that the pathways including the greatest number of genes were the “cancer pathways.” Moreover, among the miRNAs included in this research, the “Hippo signaling pathway” was shown to include the greatest amount of miRNAs (Table 3).

Figure 2 depicts the cluster analysis and heatmap of the relationship with the pathways of miRNAs that were upregulated at the greatest rate and downregulated at the lowest rate in our study, as determined by in silico analysis with the DIANA miRPath v3.0 online web server.

Figure 3 displays the heatmap linked with gene ontology (GO) analysis of miRNAs that were upregulated at the greatest relative rate and downregulated at the lowest relative rate in our study. According to our analysis, the associated miRNAs affected 6910 and 6536 genes involved in cellular components (GO:0005575) and biological processes (GO:0008150), respectively.

The connection between the miRNA and lncRNAs which were found to be upregulated at the highest rate and downregulated at the lowest rate in our study, was examined by in silico analysis performed with DIANA lncBase v3.0 online software. This analysis found a correlation between lncRNAs with the MEG3 family, PTENP1, and miRNAs detected with HAR1B. The associated miRNAs are presented in Table 4.

When the demographic data of the patients and their correlation with miRNA and lncRNAs detected in the circulation were investigated, a weak positive correlation was found between hsa-miR-122-5p and BMI among miRNAs (r_s_: 0.351; *p* < 0.05). There was no significant relationship between other parameters and miRNAs. Moreover, among the lncRNAs, a weak positive correlation was found with MALAT1 between FBS (r_s_: 0.403; *p* < 0.05), insulin (r_s_: 0.384; *p* < 0.05), HOMA-IR (r_s_: 0.298; *p* < 0.05), and HbA1c levels (r_s_: 0.306; *p* < 0.05).

## 4. Discussion

Currently, NAFLD is the most common chronic liver condition worldwide. Although it is known that NAFLD affects several organs and regulatory pathways outside of the liver, the underlying molecular processes of the illness remain obscure [14]. In this investigation, we focused on identifying the miRNAs and lncRNAs, two different families of non-coding RNAs (ncRNAs), in the serum samples of NAFLD patients, demonstrating their existence and rates in circulation and evaluating their interrelationships.

Numerous studies have been conducted on miRNAs that may have an effect on the molecular mechanism of NAFLD, and the reviews or meta-analyses that resulted from these studies reported that the expression levels of numerous miRNAs, such as miR-122, miR-34, miR-33, miR-21, miR-192, miR-375, miR-146b, miR-221, and miR-222, varied. Several studies suggest that miR-122, miR-33, miR-34a, and miR-21 regulate liver metabolism, and it has been claimed that they can be employed as invasive biomarkers [14,15]. Similarly, it has been reported that circular miRNAs can be used as biomarkers in NAFLD as well as in NASH, as these miRNAs also differentiate throughout the development to NASH [16]. Our research yielded comparable results to those of earlier studies. Specifically, the fact that certain miRNA levels varied with different grades of the disease suggested that they may be used as biomarkers if the cut-off values were determined for larger cohort groups.

The pathways of fatty acid biosynthesis and fatty acid metabolism have been linked to the development of NAFLD. It has been reported that this pathway can be used for therapeutic purposes [17]. According to our study, both upregulated and downregulated miRNAs had a significant impact on these pathways. Therefore, it has been hypothesized that silencing and stimulating these circulating miRNAs to affect these fatty acid-related pathways might serve as a therapeutic option. In addition, our research revealed that “cancer pathways” and the “Hippo signaling pathway” were significantly affected. Similar to our results, Wu et al. showed in their research that “pathways in cancer” are affected by the progression of the disease using bioinformatic techniques [18]. By regulating IRS2 expression, Jeong et al. found that the Hippo signaling system’s communication with the AKT signaling pathway protected against NAFLD in a study on a mouse model. Moreover, they reported that disruption of this pathway may contribute to the advancement of liver cancer [19]. In our study, 11 miRNAs affecting the “Hippo signaling pathway” were detected, and these were among the miRNAs that showed significant changes, supporting the study by Jeong et al. The gene ontology analysis of the miRNAs obtained in this study revealed that these miRNAs regulate genes related to cellular components and biological processes. Moreover, Jia et al. found that cellular component and biological process-related gene expressions in the microarray data of NAFLD patients are essential for understanding the molecular processes underlying NAFLD [20]. This result also confirms the conclusions which we reached in our study on miRNAs.

In mouse models, Zou et al. demonstrated that lncRNA MEG3 (maternally expressed gene 3) upregulates the sirtuin 6 (SIRT6) and enhancer of zeste homolog 2 (EZH2) genes, which regulate fat accumulation, inflammation, and the development of NAFLD. In addition, they stated that MEG3 should be increased for NAFLD control [21]. Zamani et al. found that curcumin increased MEG3 while decreasing its expression in hepatocellular carcinomas. Additionally, they reported that MEG3 exerted this effect through miRNAs [22]. Furthermore, lncRNAs such as nuclear paraspeckle assembly transcript 1 (NEAT1), hox transcript antisense RNA (HOTAIR), metastasis-associated lung adeno-carcinoma transcript 1 (MALAT1), and HOXA distal transcript antisense RNA (HOTTIP) are upregulated and play a role in the progression of liver fibrosis and hepatocellular carcinoma [23]. Similarly to previous studies, our analysis found that NEAT1, MALAT1, and MEG3 were upregulated in all samples, whereas MEG3 was downregulated. Although not detected in all samples, upregulation of HOTAIR was also observed. Nevertheless, there are contradictory data addressing the downregulation of lncRNA NEAT1 and its contribution to the correction of NAFLD in rats via the mTOR/S6K1 signaling pathway [24]. In addition, studies indicate that lncRNA PTENP1 suppresses the oncogenic PI3K/AKT pathway in hepatocellular carcinomas, and can be used to treat hepatocellular carcinomas via miRNAs such as miR-17, miR-19b, and miR-20 [25]. Similarly, lncRNA PTENP1, which is believed to be involved in a complicated autophagy process in the liver, was downregulated in all NAFLD patients in our study [26].

Matboli et al. [6] reported that miRNAs (mir-650 and miR-1205) and lncRNAs (RPARP-AS1 and SRD5A3-AS1) interact in the pathogenesis of NAFLD. In our study, in silico analysis showed that there may be more miRNA–lncRNA interactions in addition to this one.

While there have been studies demonstrating that circulating miRNAs or lncRNAs were correlated with many parameters in metabolic diseases such as NAFLD [27,28], there was also research demonstrating that they were not correlated with these parameters [29]. In our investigation, only circulating hsa-miR-122-5p and lncRNA MALAT1 levels were observed to have a weak connection with certain parameters.

Our study’s limitations included its single-center design and the absence of biopsies and histopathologies in grade evaluations of NAFLD patients. Ultrasound examinations were used in our study instead of an invasive procedure such as a biopsy, which was considered sufficient for evaluating the data obtained from patients. RNA-Seq methods, which can discover novel miRNAs and lncRNAs, were not utilized in our study, which was another limitation. We used commercial qPCR profiler kits that were comparatively less expensive than RNA-Seq, and our approach supplied comprehensive data without the requirement for qPCR data validation, which is necessary in methods such as RNA-Seq and micro-array.

## 5. Conclusions

In this study, in vitro and in silico miRNA and lncRNA interactions in NAFLD patients were investigated. Compared to healthy controls, in NAFLD patients, miR-21 was upregulated by about 2-fold and LncRNA NEAT1 by 2.9-fold, whereas miR-197 and lncRNA MEG3 were downregulated by 0.25- and 0.41-fold, respectively. In addition, while lncRNA MEG3 and PTENP1 interacted with distinct miRNAs, miRPath analysis demonstrated that the fatty acid biosynthesis pathway was affected. It is evident that the interaction between lncRNAs and miRNAs is involved in the molecular pathways underlying the development of NAFLD. In light of our results, we recommend that thorough multicenter clinical research on the silencing and stimulating miRNAs and lncRNAs discovered in this interaction should be designed.

## Figures and Tables

**Figure 1 biomedicines-11-00574-f001:**
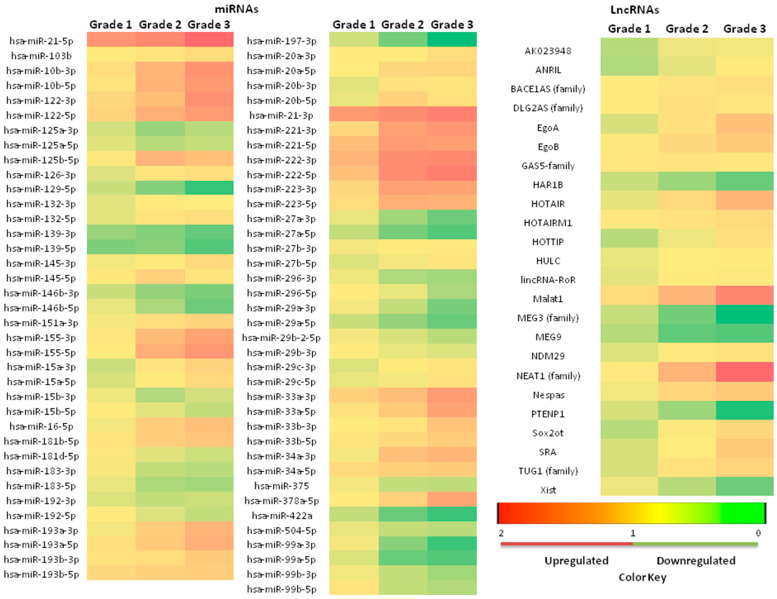
Heatmap of statistically significant miRNA and lncRNA expression profiles detected by qPCR in NAFLD patients compared to the healthy controls.

**Figure 2 biomedicines-11-00574-f002:**
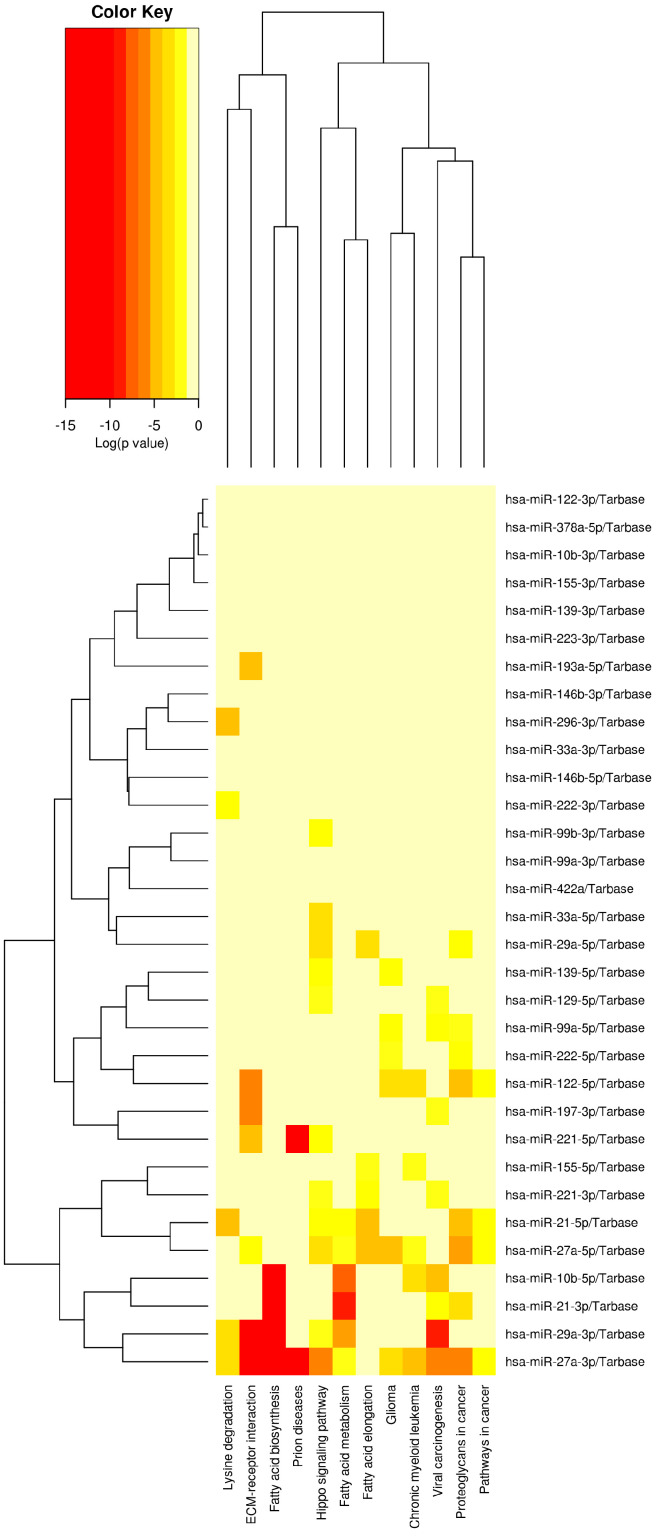
Cluster analysis and the heatmap of the relationship with the pathways of miRNAs.

**Figure 3 biomedicines-11-00574-f003:**
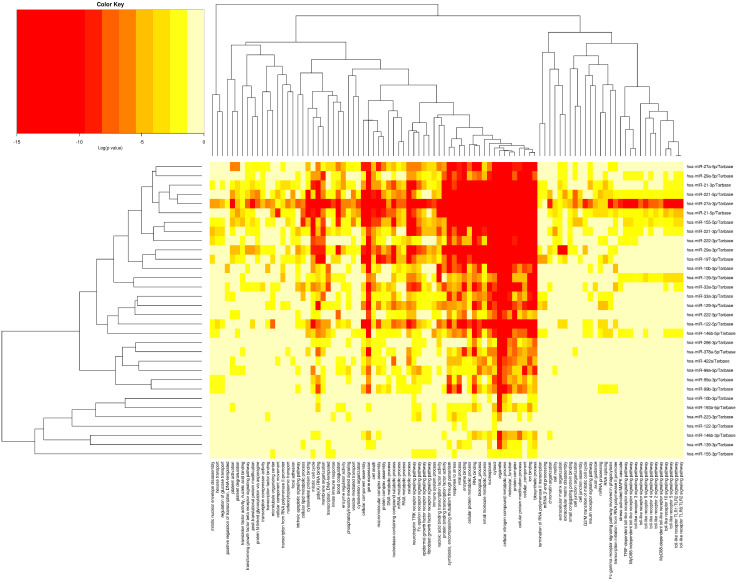
The heatmap associated with gene ontology (GO) analysis of miRNAs.

**Table 1 biomedicines-11-00574-t001:** Distribution of demographic data in NAFLD patients and healthy controls.

Data	NAFLD (n: 180) Mean ± SD	Healthy Controls (n: 60) Mean ± SD	*p* *
Gender (Female/Male)	105/75	30/30	
Age (years)	39.09 ± 11.91	38.82 ± 8.57	*p*: 0.853
HOMA_IR index	4.49 ± 2.53	1.53 ± 0.79	*p* < 0.001
FBS (mg/dL)	106.34 ± 11.53	83.25 ± 9.91	*p*: 0.003
PPBS (mg/dL)	113.18 ± 20.91	105.10 ± 14.40	*p*: 0.912
HbA1c (%)	5.59 ± 0.60	5.07 ± 0.28	*p*: 0.897
Insulin (mIU/L)	17.52 ± 8.29	7.23 ± 2.72	*p* < 0.001
ALT (U/L)	35.32 ± 18.70	24.75 ± 10.64	*p*: 0.032
AST (U/L)	23.03 ± 8.85	14.00 ± 5.92	*p*: 0.028
TG (mg/dL)	154.43 ± 77.94	111.17 ± 25.87	*p* < 0.001
HDL (mg/dL)	49.94 ± 29.04	52.75 ± 14.60	*p*: 0.037
LDL (mg/dL)	125.83 ± 37.49	87.50 ± 19.34	*p* < 0.001
BMI (kg/m^2^)	33.45 ± 5.26	21.94 ± 2.54	*p* < 0.001

* Mann–Whitney U test, HOMA-IR: Homeostasis Model Assessment of Insulin Resistance, FBS: fasting blood sugar, PPBS: post-prandial blood sugar, ALT: alanine aminotransferase, AST: aspartate aminotransferase, TG: triglyceride, HDL: high-density lipoprotein, LDL: low-density lipoprotein, BMI: body mass index.

**Table 2 biomedicines-11-00574-t002:** Distribution of demographic data of NAFLD patient groups, separated by grade.

Data	NAFLD (n: 60) (Grade 3) Mean ± SD	NAFLD (n: 60) (Grade 2) Mean ± SD	NAFLD (n: 60) (Grade 1) Mean ± SD	Healthy Controls (n: 60) (Grade 0) Mean ± SD	*p* *
Gender (Female/Male)	35/25	35/25	35/25	30/30	
Age (years)	40.60 ± 4.93	41.64 ± 11.22	37.17 ± 10.30	38.82 ± 8.57	*p*: 0.741
HOMA-IR index	4.82 ± 2.07	4.67 ± 2.58	3.95 ± 2.30	1.53 ± 0.79	*p*: 0.013
FBS (mg/dL)	113.40 ± 7.37	106.95 ± 17.35	98.67 ± 9.86	83.25 ± 9.91	*p*: 0.016
PPBS (mg/dL)	126.50 ± 10.61	121.47 ± 37.55	91.54 ± 14.58	105.10 ± 14.40	*p*: 0.019
HbA1c (%)	5.64 ± 0.22	5.75 ± 0.59	5.24 ± 0.26	5.07 ± 0.28	*p*: 0.042
Insulin (mIU/L)	18.92 ± 8.42	17.43 ± 7.24	16.21 ± 9.21	7.23 ± 2.72	*p*: 0.018
ALT (U/L)	38.75 ± 15.65	40.73 ± 22.19	26.47 ± 13.94	24.75 ± 10.64	*p*: 0.032
AST (U/L)	23.25 ± 9.84	26.18 ± 11.52	19.71 ± 5.18	14.00 ± 5.92	*p*: 0.039
TG (mg/dL)	148.00 ± 27.26	164.50 ± 105.27	151.35 ± 101.30	111.17 ± 25.87	*p*: 0.019
HDL (mg/dL)	41.67 ± 3.06	53.98 ± 37.95	44.12 ± 14.78	52.75 ± 14.60	*p*: 0.038
LDL (mg/dL)	136.75 ± 41.02	130.93 ± 38.11	109.82 ± 28.41	87.50 ± 19.34	*p* < 0.001
BMI (kg/m^2^)	35.75 ± 7.21	33.70 ± 4.61	30.89 ± 3.17	21.94 ± 2.54	*p* < 0.001

* Kruskal–Wallis test, HOMA-IR: Homeostasis Model Assessment of Insulin Resistance, FBS: fasting blood sugar, PPBS: post-prandial blood sugar, ALT: alanine aminotransferase, AST: aspartate aminotransferase, TG: triglyceride, HDL: high-density lipoprotein, LDL: low-density lipoprotein, BMI: body mass index.

**Table 3 biomedicines-11-00574-t003:** Interaction of upregulated and downregulated miRNAs with pathways after in silico analysis with DIANA miRPath v3.0 online software.

No	KEGG Pathway	Genes	miRNAs	*p*-Value
1	Fatty acid biosynthesis (hsa00061)	3	4	<1 × 10^−325^
2	ECM–receptor interaction (hsa04512)	42	7	<1 × 10^−325^
3	Prion diseases (hsa05020)	12	2	1.22 × 10^−9^
4	Proteoglycans in cancer (hsa05205)	113	8	3.55 × 10^−8^
5	Hippo signaling pathway (hsa04390)	79	11	5.40 × 10^−5^
6	Viral carcinogenesis (hsa05203)	102	8	3.89 × 10^−3^
7	Fatty acid metabolism (hsa01212)	18	6	2.63 × 10^−1^
8	Lysine degradation (hsa00310)	24	5	8.05 × 10^−1^
9	Glioma (hsa05214)	40	6	0.0006249875
10	Chronic myeloid leukemia (hsa05220)	47	5	0.007493456
11	Pathways in cancer (hsa05200)	158	4	0.01888861
12	Fatty acid elongation (hsa00062)	9	5	0.02483997

**Table 4 biomedicines-11-00574-t004:** Distribution of miRNAs associated with lncRNA with DIANA lncBase v3.0 online software.

lncRNAs	MEG3	PTENP1	HAR1B
miRNAs	hsa-miR-10b-5p	hsa-miR-21-5p	hsa-miR-33a-5p
	hsa-miR-122-5p	hsa-miR-27a-3p	
	hsa-miR-139-5p	hsa-miR-33a-3p	
	hsa-miR-146b-3p	hsa-miR-129-5p	
	hsa-miR-21-5p	hsa-miR-221-3p	
	hsa-miR-222-5p	hsa-miR-222-3p	
	hsa-miR-27a-3p		
	hsa-miR-296-3p		
	hsa-miR-29a-3p		
	hsa-miR-33a-5p		
	hsa-miR-378a-5p		
	hsa-miR-99a-3p		
	hsa-miR-99b-3p		

## Data Availability

No new data were created.

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
