# Peer review of "Could Long Non-Coding RNA MEG3 and PTENP1 Interact with miR-21 in the Pathogenesis of Non-Alcoholic Fatty Liver Disease?"

_biomedicines, 2023, doi:10.3390/biomedicines11020574_

Round 1
Reviewer 1 Report
The paper presents the transcriptome profiling of serum miRNAs and lncRNAs expression of 180 NAFLD patients and 60 healthy controls. Candidate gene approach was taken and differentially expressed miRNAs and lncRNAs were identified between NAFLD and control subjects.
However, several issues should be resolved:
1. The paper would certainly profit from native English language editing. Some parts of the manuscript are difficult to read and hard to understand (e.g. introduction and discussion section). Pg1. Line33. Pg.2, line50, and many more.
2. In the material and method section, if serum sample was used for RNA extraction. limited RNA can be isolated from non-cell portion of blood (normally in the picogram range). However, in pg2, line79, it says 25ng of isolated miRNA, in line95, it says 25ng of RNA were used for quantification. What is the initial volume of whole blood? What type of tube was used for blood collection? Please make sure it was plasma with WBC or serum was used for total RNA isolation.
3. Could author analyze the correlation between the circulating RNA and hepatic RNA expression of identified DEmiR or LncRNA based on the published data? Such a knowledge will largely enhanced the understanding and application of these biomarker in diagnosis
4. Any explanation of some genes only express in some patients but not the rest. Does this related to RNA quality, or correlated with disease condition.
5. Legends of Fig1. 2, 3 which located on the side of heatmap are not legible
6. Data presentation can be improved
7. NAFLD patients were not used alcohol (<210 g/week for males and <140 g/week for females) was stated. However, the amount is about 17 drinks (12g alcohol is one drink)/ week for man, and almost 12 drinks/week for woman. Please clarify the alcohol consumption level of NAFLD patient.
Author Response
Dear Editor and Reviewer,
Thank you very much for your valuable comments and contribution. all comments have been carefully checked and edited on the manuscript.
You can also see our response below.
Thank you, Regards
Assoc. Prof. Mehmet Demirci
Response to Reviewer 1:
The paper presents the transcriptome profiling of serum miRNAs and lncRNAs expression of 180 NAFLD patients and 60 healthy controls. Candidate gene approach was taken and differentially expressed miRNAs and lncRNAs were identified between NAFLD and control subjects.
However, several issues should be resolved:
- The paper would certainly profit from native English language editing. Some parts of the manuscript are difficult to read and hard to understand (e.g. introduction and discussion section). Pg1. Line33. Pg.2, line50, and many more.
Response 1: Thank you for your valuable comments and contributions. Manuscript was checked by native speaker at the university editing service and relevant corrections were made. We can also send editing certificate.
- In the material and method section, if serum sample was used for RNA extraction. limited RNA can be isolated from non-cell portion of blood (normally in the picogram range). However, in pg2, line79, it says 25ng of isolated miRNA, in line95, it says 25ng of RNA were used for quantification. What is the initial volume of whole blood? What type of tube was used for blood collection? Please make sure it was plasma with WBC or serum was used for total RNA isolation.
Response 2: Thank you for your valuable comments and contributions. In our study, miRNA and total RNA isolations were performed using 200 µl serum samples. In the literature, it is reported that miRNA and lncRNA isolation studies can be performed from both serum samples and whole blood samples, but the sample type should be specified. it is stated that this difference may cause problems especially in comparisons of data in the literature. in our study, serum samples were used and RNA isolations were performed quickly. It was realized that there was an inadvertent error in the concentration and the unit was 25 ng/mL. therefore the concentration was changed to 25 pg/µl.
- Could author analyze the correlation between the circulating RNA and hepatic RNA expression of identified DEmiR or LncRNA based on the published data? Such a knowledge will largely enhanced the understanding and application of these biomarker in diagnosis
Response 3: Thank you for your valuable comments and contributions. In our study, DIANA web server was used for this purpose and both miRNA interactions and lncRNA-miRNA interaction were analyzed. these data were tried to be compared with the results in the literature.
- Any explanation of some genes only express in some patients but not the rest. Does this related to RNA quality, or correlated with disease condition.
Response 4: Thank you for your valuable comments and contributions. In our study, all miRNA expressions were detected, but the significant ones were reported. on the other hand, similar to the literature, some genes were detected in lncRNA, while some genes were not detected. this was thought to be related to lncRNAs.
- Legends of Fig1. 2, 3 which located on the side of heatmap are not legible
Response 5: Thank you for your valuable comments and contributions. legends are legible in the original version of the Figures, but this is how it looks in the word software draft.
- Data presentation can be improved
Response 6: Thank you for your valuable comments and contributions. Data presentation has been tried to be improved.
- NAFLD patients were not used alcohol (<210 g/week for males and <140 g/week for females) was stated. However, the amount is about 17 drinks (12g alcohol is one drink)/ week for man, and almost 12 drinks/week for woman. Please clarify the alcohol consumption level of NAFLD patient.
Response 7: Thank you for your valuable comments and contributions. Thank you for the very important warning. all our patients are Muslim patients and do not consume alcohol. this statement has been edited for this reason.
Reviewer 2 Report
- Write abbreviated form of NAFLD in the abstract as NAFLD was used more than 1 time.
- Please highlights the results in the abstract.
- Even Conclusion was not clear in the abstract.
- Please add the latest reference for describing NAFLD in the introduction (https://pubmed.ncbi.nlm.nih.gov/36557278/)
- Authors need to describe the relation between NAFLD and miRNA in the introduction
- On what basis 180 NAFLD subjects and 60 controls were included?
- Write clearly about inclusion and exclusion criteria of NAFLD cases and controls
- Please add about quantification details of genomic DNA
- Define P value for age in table 1 and in results section
- Write accurate P values in Table 1 and 2 instead of (p<0.05 and p>0.05)
- In Discussion, authors can delete full form of NAFLD
- Authors can correlate the genotype data with any of the variable applied in this study.
- Please omit the sentence as “Our study was the first” throughout the manuscript.
Author Response
Dear Editor and Reviewer,
Thank you very much for your valuable comments and contribution. all comments have been carefully checked and edited on the manuscript.
You can also see our response below.
Thank you, Regards
Assoc. Prof. Mehmet Demirci
Response to Reviewer 2:
- Write abbreviated form of NAFLD in the abstract as NAFLD was used more than 1 time.
Response 1: Thank you for your valuable comments and contributions. The correction was carried out in the abstract.
- Please highlights the results in the abstract.
Response 1: Thank you for your valuable comments and contributions. The correction was carried out in the abstract.
- Even Conclusion was not clear in the abstract.
Response 3: Thank you for your valuable comments and contributions. The correction was carried out in the abstract.
- Please add the latest reference for describing NAFLD in the introduction (https://pubmed.ncbi.nlm.nih.gov/36557278/)
Response 4: Thank you for your valuable comments and contributions. The reference added and all reference number changed.
- Authors need to describe the relation between NAFLD and miRNA in the introduction
Response 5: Thank you for your valuable comments and contributions. the relation between NAFLD and miRNA describtion added in the introduction.
- On what basis 180 NAFLD subjects and 60 controls were included?
Response 6: Thank you for your valuable comments and contributions. Consecutive patients and control randomly included in this study. Due to the fact that the kits used in our study are very expensive and contain plates for a certain number of samples, the numbers were determined based on budget adaptation and this number of samples could be studied.
- Write clearly about inclusion and exclusion criteria of NAFLD cases and controls
Response 7: Thank you for your valuable comments and contributions. inclusion and exclusion criteria were added.
- Please add about quantification details of genomic DNA
Response 8: Thank you for your valuable comments and contributions. genomic DNA contamination protocol added.
- Define P value for age in table 1 and in results section
Response 9: Thank you for your valuable comments and contributions. P values added in the tables
- Write accurate P values in Table 1 and 2 instead of (p<0.05 and p>0.05)
Response 10: Thank you for your valuable comments and contributions. P values added in the tables
- In Discussion, authors can delete full form of NAFLD
Response 11: Thank you for your valuable comments and contributions. full form of NAFLD delete in the discussion
- Authors can correlate the genotype data with any of the variable applied in this study.
Response 12: Thank you for your valuable comments and contributions. Correlations were performed between expression and other data. Can see in the result section; a weak positive correlation was found between hsa-miR-122-5p and BMI among miRNAs (rs: 0.351; p<0.05). There was no significant relationship between other parameters and miRNAs. Moreover, among the lncRNAs, a weak positive correlation was found with MALAT1 between FBS (rs: 0.403; p<0.05), insulin (rs: 0.384; p<0.05), HOMA-IR (rs: 0.298; p<0.05) and HbA1c levels (rs: 0.306; p<0.05).
- Please omit the sentence as “Our study was the first” throughout the manuscript.
Response 13: Thank you for your valuable comments and contributions. the sentence was deleted.
Round 2
Reviewer 2 Report
Authors have justified and can be accepted in the present form
Author Response
Dear Reviewer,
Thank you very much for your valuable comments and contribution.
Thank you, Regards
Assoc. Prof. Mehmet Demirci
Response to Reviewer
Authors have justified and can be accepted in the present form
Response: Thank you very much for your valuable comments and contribution.